# Battery Surface and Edge Defect Inspection Based on Sub-Regional Gaussian and Moving Average Filter

**Haibing Hu** *[ID], **Bo Zhang, Dongjian Xu and Guo Xia**

National Engineering Lab of Special Display Technology, State Key Lab of Advanced Display Technology, Academy of Opto-Electronic Technology, Hefei University of Technology, Hefei, Anhui 230009, China
* Correspondence: huhb@hfut.edu.cn; Tel.: +86-1811-097-2015

**Abstract:** Detecting the defects of a battery on the surface and edge has always been difficult, especially for concave and convex ones, thereby seriously affecting its quality. Thus, sub-regional Gaussian and moving average filtering are innovatively proposed in this study considering the effect of the nonuniform background illumination of the battery edge and the difference between the edge background and the internal surface defects of the battery. The battery surface image is divided into two areas, namely, edge area $W_1$ and inner area $W_2$. Gaussian and moving average filtering are carried out row-by-row and column-by-column in the inner area $W_2$ and the edge area $W_1$, respectively. The algorithm is tested on 600 battery samples that mainly possess concave and convex defects. The proposed method has higher detection accuracy and lower omission detection rate than the traditional unpartitioned processing method, especially in detecting the accuracy of edge defects. The accuracy rates were approximately 20% higher than that obtained by the traditional processing algorithm. The proposed method has remarkable real-time performance that can process four 8192 × 10,240 pixel battery images per second, thereby meeting the industrial production line speed requirements while satisfying accuracy. The proposed method has been applied in actual production for defect inspection.

**Keywords:** battery defects; sub-regional; Gaussian filter; moving average filter; accuracy; real-time performance

## 1. Introduction

The battery is an essential product that has been widely used in many fields, such as electronics, communication, instrument, transportation, and machinery manufacturing [1–4]. The demand for a battery with high surface, performance, and quality increases annually given the rapid development of modern science and technology. Various defects, such as scratch [5], concave, and convex, appear on the battery's surface during production due to defects of raw material, rolling equipment, and processing technology [6,7]. These defects would not only affect the battery's appearance, but also the product's quality and performance. The battery's surface defect inspection is repetitive and needs tremendous concentration. Traditional inspection methods, including artificial visual [8], frequency flight [9], infrared [10], magnetic flux leakage [11], and ultrasonic [12], have disadvantages, such as being time consuming and having a high missing rate and low inspection precision due to complex and extreme environment. Among these problems, edge defect detection has always been difficult to solve because of curved edges and uneven reflections (Figure 1). Thus, the results of the traditional methods do not satisfy the requirements. In addition, companies need to store the data information of the battery's surface, especially its defect information, to revalidate its quality upon completion.

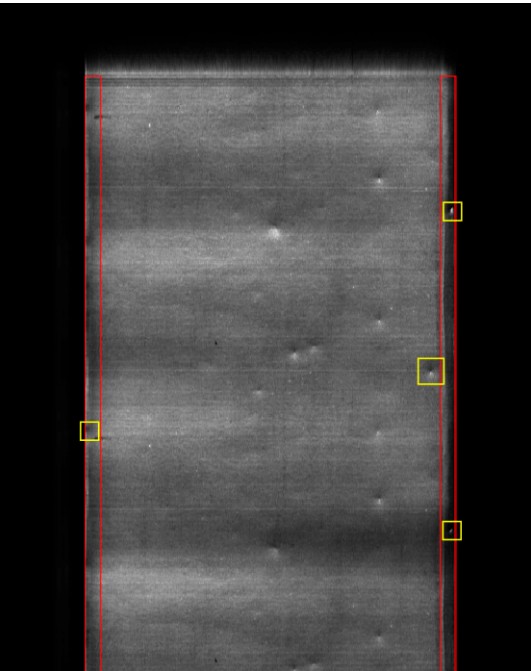

**Figure 1.** The red mark indicates the hard-to-detect area and the yellow mark indicates a defect that is not easily detected on the edge.

Computer vision has been widely used and has been proven to have advantages, such as high speed, effective, and high inspection precision [13–15]. Defects are caused by different factors and appear in various types. The core of image processing is to extract useful features from the surface images and to track and address the cause of the different types of defects. Several popular feature extraction methods can be used in image processing. Li proposed an efficient scale-invariant feature transform (SIFT) algorithm to accelerate feature extraction [16]. Bay proposed other efficient speed-up robust feature (SURF) algorithms to further accelerate feature extraction based on SIFT [17]. Ojala presented a local binary pattern (LBP) algorithm, which was a redefinition of the grayscale values of the original image and a simple combination of histograms [18]. Li presented a segmentation algorithm based on Canny edge defects on the surface of the battery cathode [19]. Tian introduced an extreme learning machine algorithm combined with a genetic algorithm for the surface defect identification of hot-rolled steel plates, which improved detection accuracy by self-learning through training samples [20]. Li used Gabor filters and the pulse coupled neural network (PCNN) to identify defects. Gabor filters are used to enhance the contrast of images captured by camera, and the defect areas are automatically segmented by PCNN with an adaptive parameter setting [21]. Li used local normalization (LN) to enhance image contrast, which was nonlinear and illumination independent, and to detect defects using the defect localization based on the projection profile (DLBP), which was robust to noise and prompt [22].

There are few articles that detect concave and convex defects of the battery we mentioned; the above image detection methods are generally used for the detection of defects on the surface of metal plates. Therefore, we proposed an efficient algorithm for battery surface and edge defect inspection based on sub-regional Gaussian and moving average filtering. The algorithm mainly identifies the concave and convex defects on the surface and batteries' edges because these are defects that affect the batteries' quality during the production process. Moreover, detecting defects on edges has always been difficult.

## 2. System Composition

### 2.1. System Structure

The system structure of the battery surface and edge defect inspection based on sub-regional Gaussian and moving average filtering mainly include linear array charge-coupled device (CCD) camera, light source, and industrial personal computer (IPC) with screen and test samples (Figure 2a). The uncertain geometrical defects on the battery's surface (especially the concave and convex defects) are the drawbacks of the battery surface defect inspection. The battery's test samples (Figure 2b) are made of metal, and uneven illumination may occur during the test. The linear array light-emitting diode (LED) [23] and linear array CCD camera [24] have been used to avoid these effects during image acquisition and defect inspection. The linear array LED light source is used to provide suitable illumination. The linear array CCD camera continuously scans the battery's surface to form a uniform 2D image that can complete image acquisition on the entire battery's surface. The image of the battery's surface collected through this approach is rectangular in shape. This shape easily and accurately segments the defects. Feature extraction is convenient, and the accuracy of detecting the battery's microdefects is improved.

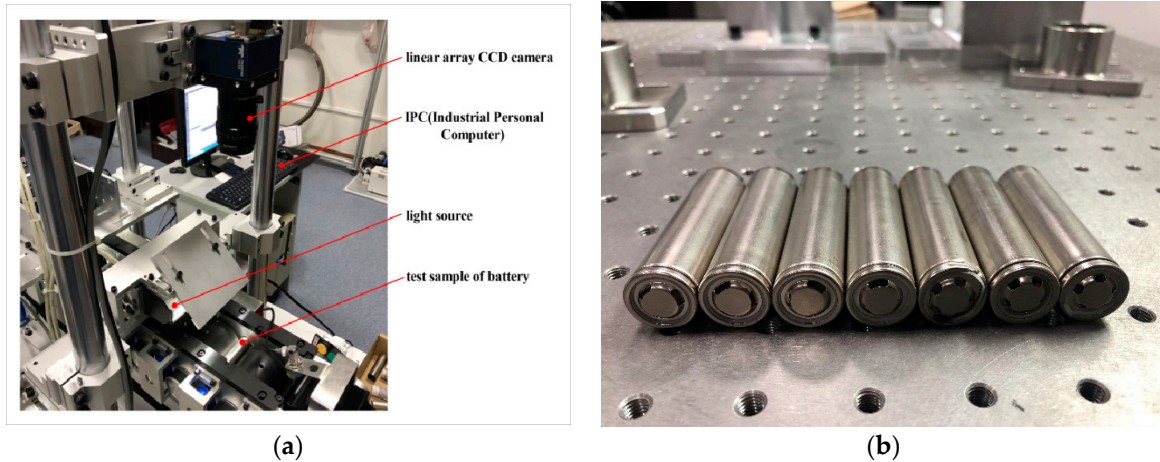

(**a**)　　　　　　　　　　　　　　　　　　　　　　　　　(**b**)

**Figure 2.** System structure. (**a**) Detection system; (**b**) Test samples of battery.

The linear array CCD camera has higher inspection precision and lower cost than the face array CCD camera when inspecting images with the same size and resolution. The linear array CCD camera has high scanning speed, high frequency response, and can realize dynamic measurement and work under low illumination intensity.

The images are acquired using one $2048 \times 1$ pixel linear array CCD camera produced by Teledyne DALSA. The pixel size of the camera is 7.04 μm × 7.04 μm, the maximum line frequency can reach up to 80 KHz, and sensitivity is $320 \frac{\text{DN}}{\left(\frac{\text{nJ}}{\text{cm}^2}\right)}$, 12 bit, $1 \times gain$, and supports multiple regions of interest (ROI) functions.

### 2.2. Software Structure

Image processing methods are the core components of this study, which are divided into several sections. Figure 3 shows the proposed software structure, including image acquisition, ROI extraction, median filtering, sub-regional Gaussian and moving average filtering, threshold setting, binary large object (blob) analysis, mainly about noise reduction and operation, analysis of defect characteristics, and defects marked on the original image.

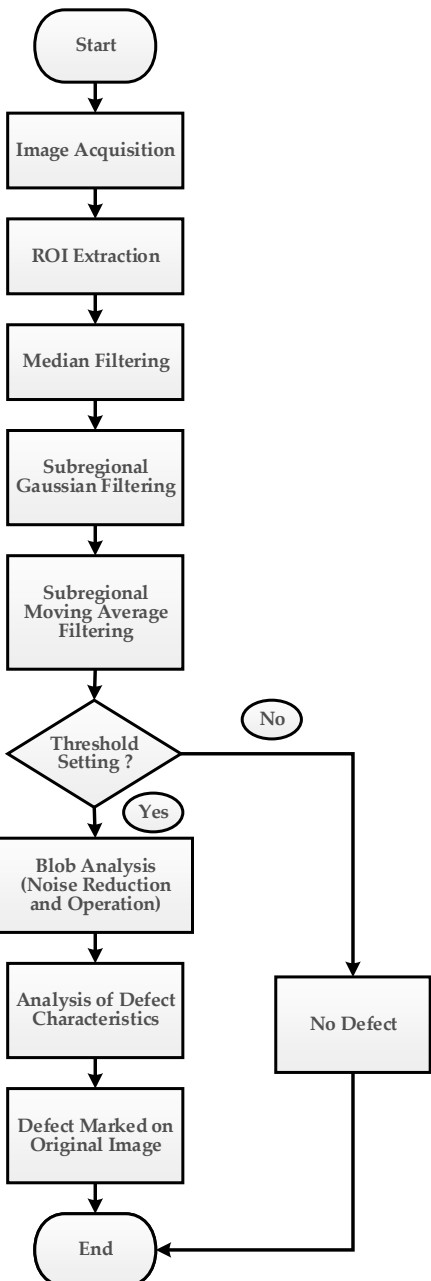

**Figure 3.** Software structure.

## 3. Principle of Defect Inspection Algorithm

### 3.1. ROI Extraction

Region of interest (ROI) extraction is the first step of image processing. The original image captured by the linear CCD array camera is a constant 2D image (Figure 4a). The size of the original acquired image is 8192 × 10, 240 pixel according to the parameter settings of the linear CCD array camera used. The original image also includes invalid areas. Determining the vertical line that best fits the ROI position of the battery edge in the image is necessary because of the battery's constant width and the different luminance between the background and the battery surface. The specific ROI extraction processing can be described as follows:

1. The pixel value $px_i$ of each line is identified, and the maximum value $px_{max}$ is obtained. The pixel value of each point is divided by the maximum value $\frac{px_i}{px_{max}}$. Correspondingly, the same procedure is performed for column $\frac{py_i}{py_{max}}$.

2. The pixel value between ROI and the background dramatically changes. The boundary point $px_0 = \frac{1}{n}\sum\limits_{i=1}^{n}\frac{px_i}{px_{max}}, py_0 = \frac{1}{m}\sum\limits_{i=1}^{m}\frac{py_i}{py_{max}}$ is determined using this feature, where n is the total number of rows in the image, and m is the total number of columns.

3. The column width of ROI was extracted from left to right (Figure 4b).

4. The row width of ROI was extracted from top to bottom (Figure 4c).

5. ROI extraction is completed, and ROI is divided into two sections, namely, edge part $W_1$ and inner part $W_2$ (Figure 4d).

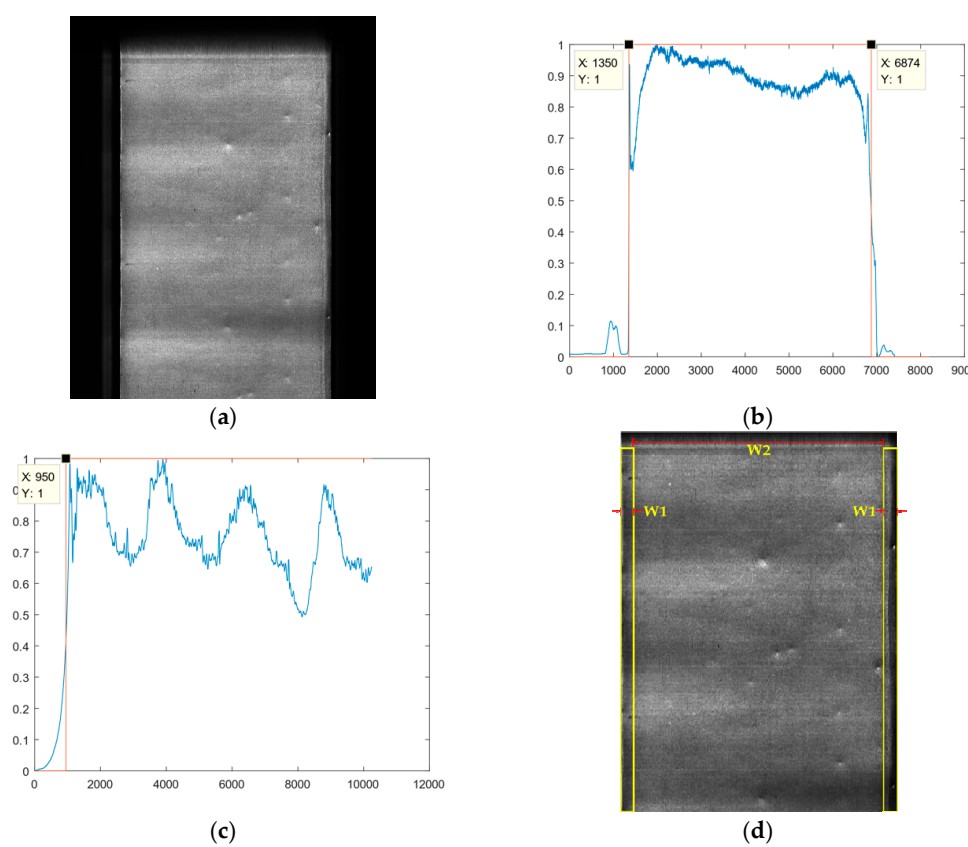

**Figure 4.** Processing of Region of interest (ROI) extraction. (**a**) Original image ($8192 \times 10,240$ pixels); (**b**) ROI extracted of columns; (**c**) ROI extracted of rows; (**d**) ROI extraction finished ($5525 \times 9291$ pixels).

After ROI extraction, a series of image processing algorithms, including Gaussian filtering [25], moving average filtering (MAF) [26], blob analysis [27], and defect characteristic analysis [28], are applied. The sub-regional processing method is creatively proposed on the basis of Gaussian filtering and MAF considering the effect of nonuniform background illumination of the battery edge and the difference between the edge background and the internal surface defects of the battery. Gaussian filtering and MAF are carried out row-by-row in the inner area $W_2$ and column-by-column in the edge area $W_1$. This approach is different from most module or global processing methods. The proposed method has higher defect detection accuracy and lower omission detection rate than the traditional unpartitioned processing methods. The detailed principles of the proposed algorithm are described in

the following section. To prove the advancement of our proposed algorithm, we compared it with the traditional unpartitioned processing.

## 3.2. Gaussian Filtering

Enhancing image quality is necessary because the images from the image acquisition module contain noise. Row-by-row Gaussian filtering is applied to filter image noises according to the speed and performance of image-enhancing processes. The 1D digital Gaussian filter can be expressed as Equation (1). In noise filtering, a model is commonly assumed as the convolution of two signals (Equation (2)).

$$G(x) = \frac{1}{\sqrt{2\pi}\sigma} e^{-\frac{x^2}{2\sigma^2}} \tag{1}$$

$$M(x) = S(x) \otimes Z'(x) \tag{2}$$

where $\sigma$ represents the standard deviation of the Gaussian filter, $x$ is the pixel index, $M(x)$ is the observed signal, $S(x)$ is the original signal, $Z'(x)$ is the identical distribution Gaussian noise with zero mean, and $\sigma^2$ is the variance.

The partition Gaussian process is applied, considering the effect of the nonuniform background illumination of the battery edge and the difference between the edge background and the internal surface defects of the battery. Gaussian filtering is processed row-by-row in the inner area $W_2$ and column-by-column in the edge area $W_1$. Figure 5 shows the results of partition Gaussian method and traditional Gaussian methods.

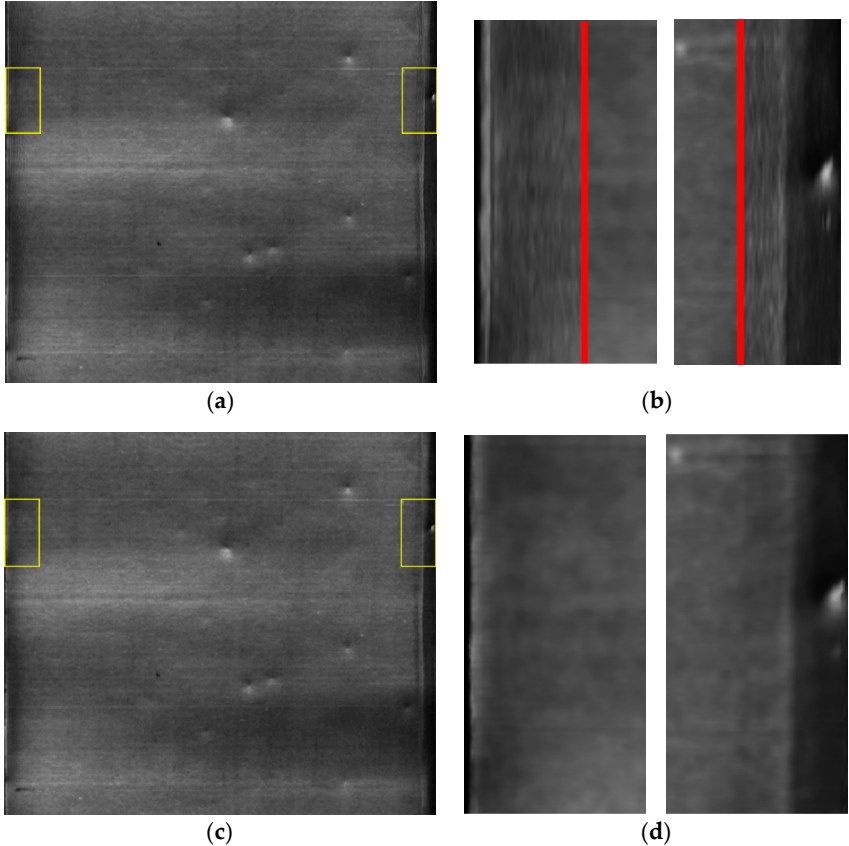

**Figure 5.** Results of Gaussian filtering. (**a**) Result of partition Gaussian processing; (**b**) Partial magnification result of internal horizontal filtering of image while edge longitudinal filtering; (**c**) Result of traditional Gaussian processing without partition processing; (**d**) Partial magnification result of image global lateral filtering.

### 3.3. Moving Average Filtering (MAF)

Figure 6 shows the grayscale variation on the surface and edge of the battery. We proposed to find a grayscale trend curve by analyzing the grayscale curve. The defect position is highlighted, and the gray background is weakened through subtraction. Therefore, we used these features to smoothen the curves, perform subtraction, highlight the defects, and set an adaptive threshold to identify the defects.

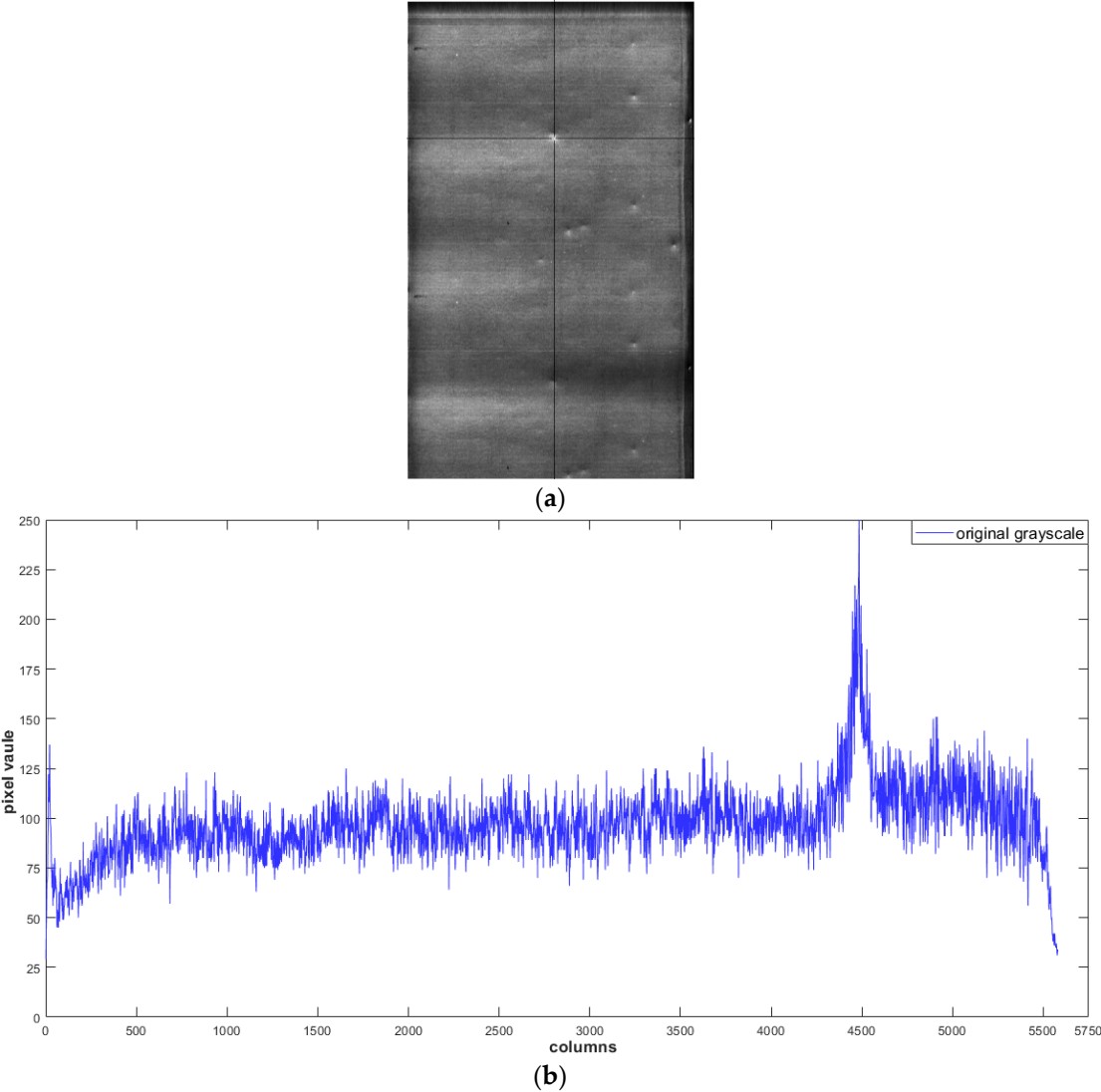

**Figure 6.** The grayscale variation of original ROI image of the battery. (**a**) ROI; (**b**) The grayscale variation on a row of the ROI image.

MAF is defined by averaging a number of points from the input signal to produce each point in the output signal. MAF can be regarded as a window of a certain size (N, in this case) that moves along the array that constitutes from the input signal, one element at a time (Figure 7). The average of all elements in the current window correspond to the MAF output.

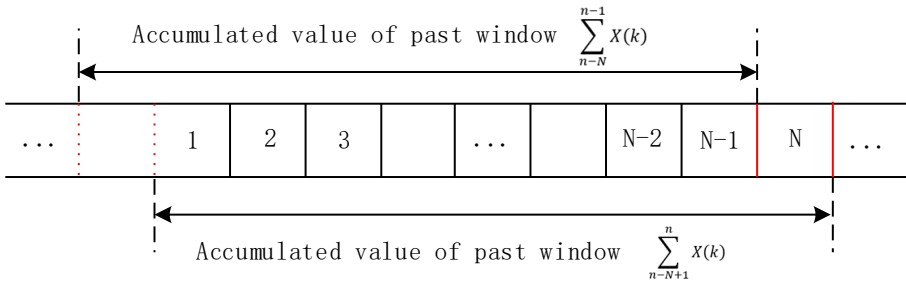

**Figure 7.** Principle of moving average filtering.

The relationship between the signal of the input and the output of the MAF can be clearly described in Equations (3) and (4), respectively:

$$\sum_{n-(N-1)}^{n} X(k) = \sum_{n-N}^{n-1} X(k) - X(n-N) + X(n) \tag{3}$$

$$Y(n) = \frac{1}{n} \sum_{n-(N-1)}^{n} X(k) \tag{4}$$

where $X(n)$ and $Y(n)$ are the pixel values of the input and output signals of the MAF, respectively, and N is the size of the moving window, that is, the number of samples of the input pixel per moving period.

Integrating Equation (3) into Equation (4), the transfer function from the MAF input to its output in the time-domain can be described in Equation (5) as follows:

$$Y(n) = Y(n-1) + \frac{X(n) - X(n-N)}{N} \tag{5}$$

The transfer function indicates that MAF is a finite impulse response (FIR) filter, which has linear phase characteristics and good robustness.

Figure 8 illustrates the results of MAF. Figure 8a shows the MAF curve plots with different numbers of samples, whereas Figure 8b shows the residual changes correspondingly. MAF1 has 64 samples ($N = 64$), MAF2 has 256 samples ($N = 256$), MAF3 has 512 samples ($N = 512$), MAF4 has 1024 samples ($N = 1024$), and MAF5 has 2048 samples ($N = 2048$) within a moving period. The analysis of the curve changes indicates that the residual is prominent when the number of samples ($N$) is large. However, $N = 512$ is selected as the final choice of the experiment considering the computation time and the effect of the harmonics. The appropriate threshold is set on the basis of the characteristics of the normal distribution to extract the defects, and the threshold is set to dynamic threshold in our experiment in accordance with Equation (6). Figure 8c illustrates the result of defect extraction when MAF has 512 samples ($N = 512$).

$$I_{thr} = \mu \pm 3\sigma \tag{6}$$

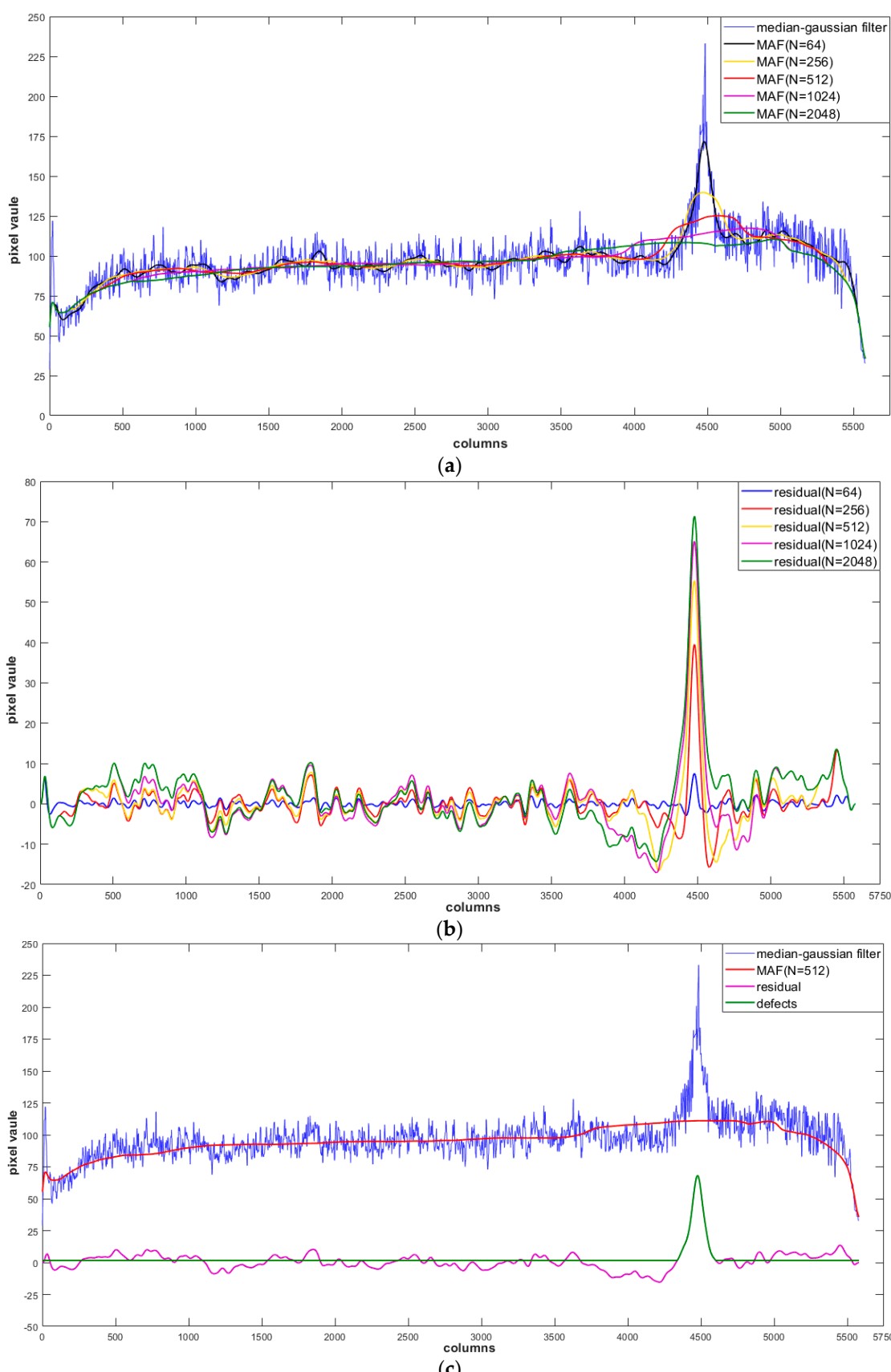

**Figure 8.** Results of moving average filtering (MAF). (**a**) Result of MAF curve plots with different numbers of samples; (**b**) Result of residuals with different numbers of samples; (**c**) Result of defects extraction when MAF has 512 samples (*N* = 512).

Figure 9 shows the results of partition processing on the image and the traditional processing without partition, where MAF is processed row-by-row in the inner area $W_2$ and column-by-column in the edge area $W_1$.

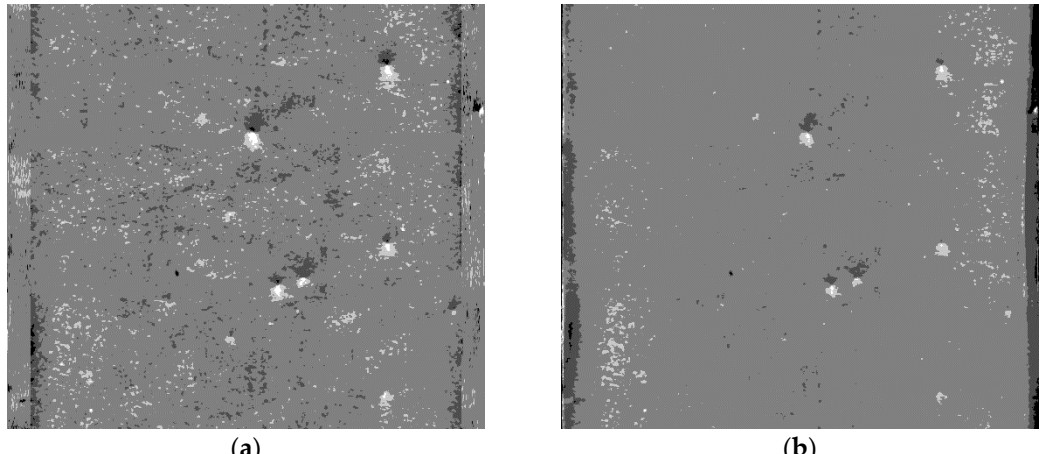

(**a**)                                        (**b**)

**Figure 9.** Results of defects extraction on image. (**a**) Result of partition defects extraction when MAF has 512 samples ($N = 512$); (**b**) Result of traditional defects extraction without partition processing.

### 3.4. Blob Analysis

In this section, blob analysis is mainly applied for noise cancellation and area feature extraction. In computer vision, blob refers to a connected area composed of features, such as similar colors and textures in an image. In a word, blob analysis refers to the geometric analysis of a connected area to obtain important geometric features, such as area, center point coordinates, centroid coordinates, minimum circumscribed rectangle, and spindle. It can separate the target from the background and can calculate the target number, location, shape, orientation, and size, as well as the topology among related spots.

Figure 10 shows the use of open operation to separate the defect from the background and to remove some of the noise. Figure 11 shows the area extraction.

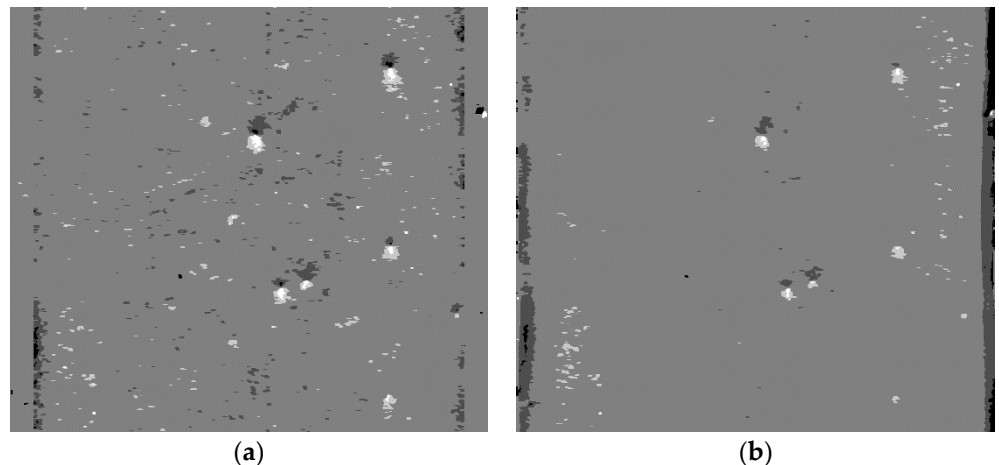

(**a**)                                        (**b**)

**Figure 10.** Results of noise reduction. (**a**) Result of partition noise reduction; (**b**) Result of traditional noise reduction without partition processing.

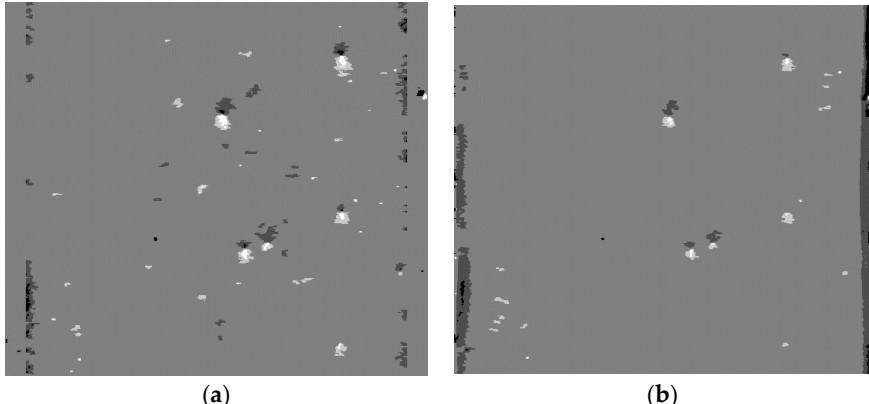

(**a**)　　　　　　　　　　(**b**)

**Figure 11.** Results of area extraction. (**a**) Result of partition area extraction; (**b**) Result of traditional area extraction without partition processing.

### 3.5. Analysis of Defect Characteristics

A bright and dark defect with a certain symmetry relationship is formed on the image due to the nonuniformity of the reflected light of the concave and convex defects, and boundary search is performed on the image after blob analysis based on the characteristics of defects. Figure 12 shows the defects marked on the original image after the boundary search of defects. The figure also shows that the partition processing has higher accuracy than the traditional processing without partition. This difference is especially reflected in the edge defect detection.

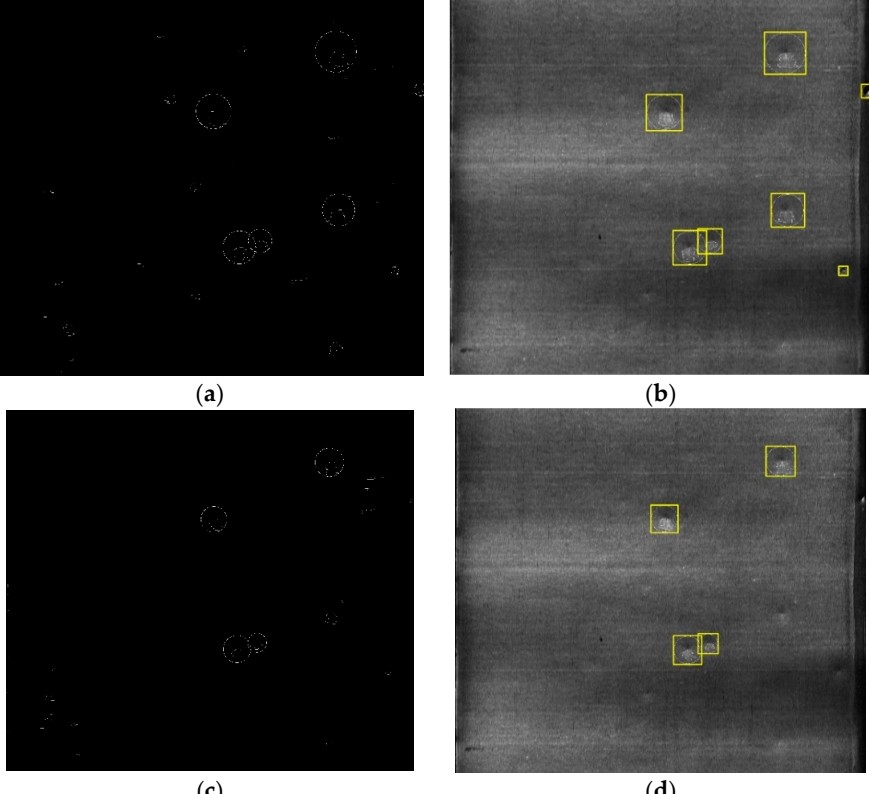

(**a**)　　　　　　　　　　(**b**)

(**c**)　　　　　　　　　　(**d**)

**Figure 12.** The inspection results of defects for concave and convex. (**a**) Result of boundary search of defects by partition method; (**b**) Result of defects marked on original image by partition method; (**c**) Result of boundary search of defects by traditional method without partition processing; (**d**) Result of defects marked on original image by traditional method without partition processing.

## 4. Experiments and the Analysis of Results

A total of 600 battery samples with $8192 \times 10,240$ pixels are divided into 5 groups and applied in our defection system to evaluate the effectiveness of the proposed algorithm. Figure 13 shows the experimental results of the proposed processing algorithm based on sub-regional Gaussian and MAF compared with other processing methods. Tables 1 and 2 illustrate critical data to demonstrate the superiority of the proposed algorithm. Table 1 shows the comparison of partition and traditional unpartitioned detection results, it means that the traditional method is to process horizontally or vertically on the whole picture directly, while the partition processing takes into account the uneven illumination distribution of the edge of the image, the vertical processing at the edge, and the horizontal processing inside image. Table 2 illustrates the accuracy rates and detection time with different concave and convex defect inspection methods, because there is almost no research on the surface defects of the battery, battery surface images we acquired are applied into the current popular metal surface defect detection algorithms, and the performance index is analyzed. Both algorithms were developed with MATLAB (Version R2017b, MathWorks Inc., Natick, MA, USA) and Microsoft Visual Studio 2013 and operated in a computer with Windows system (CPU 3.6 GHz Intel Core i7, Memory 8 GB).

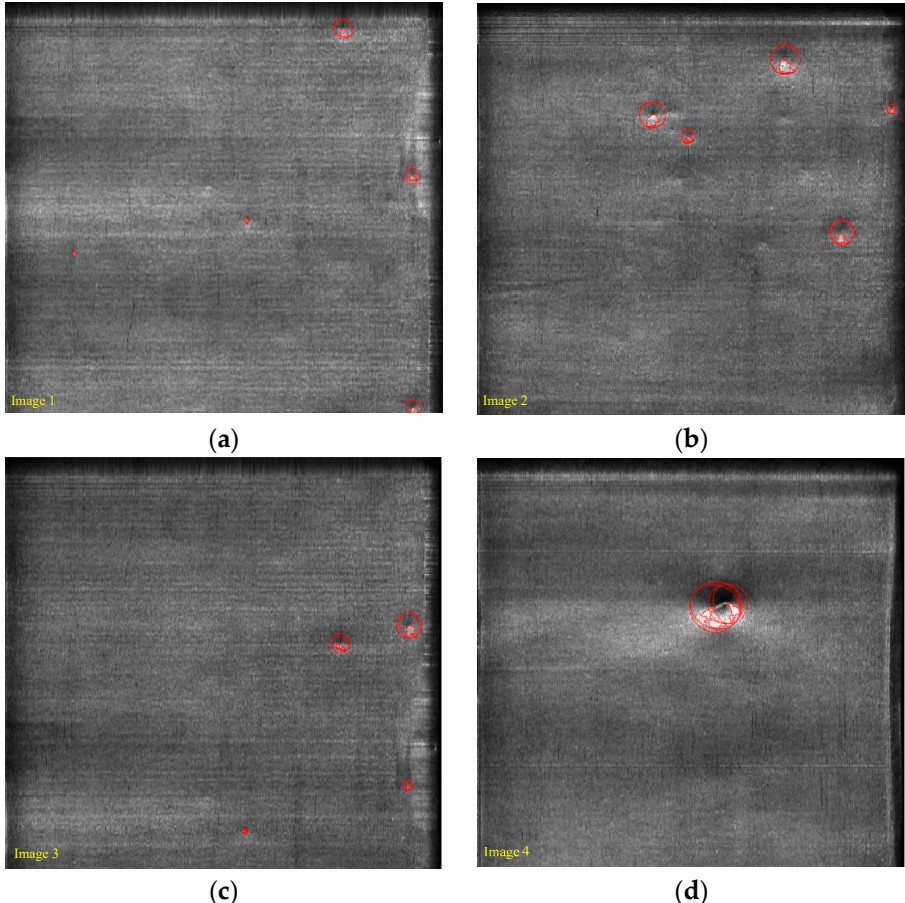

(a)   (b)

(c)   (d)

**Figure 13.** The inspection results of defects for concave and convex based on sub-regional Gaussian and MAF, where $N = 512$. (**a**) Inspection result of defects for concave and convex on image 1; (**b**) Inspection result of defects for concave and convex on image 2; (**c**) Inspection result of defects for concave and convex on image 3; (**d**) Inspection result of defects for concave and convex on image 4.

**Table 1.** Accuracy rates and detection time comparison of partition and traditional unpartitioned detection results.

| Methods | 1 | 2 | 3 | 4 | 5 | Mean | Time (s) |
|---------|-----|-----|-----|-----|-----|------|----------|
| partition | 98.82% | 98.56% | 93.54% | 96.45% | 95.35% | 95.94% | 3.3303 |
| tradition | 73.45% | 73.45% | 77.35% | 82.35% | 78.45% | 75.96% | 3.3321 |

**Table 2.** Accuracy rates and detection time with different methods.

| Methods | Defect Type | Time (s) | Accuracy (%) | Ref |
|---------|-------------|----------|--------------|-----|
| Deep Convolutional Neural Network | concave and convex | 83 (training time:133min) | 96.72% | [29] |
| LBP | concave and convex | 8.2683 | 95.13% | [30] |
| SURF | concave and convex | 7.4512 | 89.70% | [17,30] |
| Gabor-Otsu | concave and convex | 4.1859 | 82.32% | [19] |
| Polynomial Fitting | concave and convex | 3.7280 | 95.43% | Our work |
| MAF | concave and convex | 3.3303 | 95.94% | Our work |

## 5. Conclusions

This study investigates the method for battery surface and edge defect inspection, especially for the concave and convex defects that affect battery quality. Sub-regional Gaussian and MAF is innovatively proposed. The partition processing method fully considers the characteristics of uneven distribution of edge illumination, processing the image sub-region to improve the detection rate of edge defects. The proposed method has been proven to have higher defect detection accuracy and lower miss detection rate and performed very efficiently in edge defect detection compared with the traditional unpartitioned processing method which processes horizontally or vertically on the whole picture directly (Figure 13). Table 1 shows that the accuracy rates are approximately 20% higher than that obtained without the use of the partition processing algorithm. In addition, Table 2 summarizes the performances of our method and other recently reported metal surface defect inspection methods. It can be seen that our method exhibits a high accuracy and low detection time, which is comparable with the best results reported in the literature. Furthermore, our method has been applied to actual factory inspection; the detection speed required by the factory is four batteries per second, and the detection accuracy is not less than 92%. Therefore, some methods like local binary patterns (LBP), deep convolutional neural network(DCNN), and polynomial filtering do not satisfy the speed requirement although they meet accuracy requirement. The proposed method can satisfy both requirements in Visual Studio 2013, has better real-time performance and can guaranteed the accuracy. The results also show that the proposed method is effective in detecting metals surface defects and has further research significance.

**Author Contributions:** Conceptualization, B.Z. and H.H.; methodology, B.Z.; software, B.Z.; validation, B.Z., H.H. and G.X.; formal analysis, G.X.; investigation, B.Z.; resources, H.H.; data curation, D.X..; writing—original draft preparation, B.Z.; writing—review and editing, B.Z.; visualization, B.Z.; supervision, H.H.; project administration, H.H.; funding acquisition, H.H.

**Funding:** This research was funded by the National Major Scientific Instruments and Equipment Development Project of Ministry of Science and Technology of China, grant number 2013YQ220749.

**Acknowledgments:** Thanks for the technical support given from my teacher and classmates.

**Conflicts of Interest:** The authors declare no conflict of interest.

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
