# Peer review of "Battery Surface and Edge Defect Inspection Based on Sub-Regional Gaussian and Moving Average Filter"

_applsci, doi:10.3390/app9163418_

Round 1
Reviewer 1 Report
The article discusses the problem of visual inspection on the production line. The task concerns inspections for defects detection on the surface and edge of batteries. The article is clearly written, the proposed method is well described but I find no reliable comparison with other methods. It is the main criticism of the paper. The proposed method is compared only with the traditional method, without the partitioning (called by the authors "tradition", Table 1). Other methods are compared only in terms of the time of the operation (Table 2). What is the accuracy of these methods. What is more, the methods used for comparison purposes in Table 2 are rather classical one. It would be interesting to see results of performance (accuracy and speed) of contemporary methods like ones based on deep learning approach.
From minor remarks, it is worth paying attention to the diagram presented in Figure 3. There is no outgoing line from the "Noise reduction" block. What operation is performed after reduction of the noise?
I would also recommend the authors to make the database available for the public so everyone could evaluate own algorihtms and compare in future with the method presented in the paper.
Reviewer 2 Report
The author proposed a image partitioning method to be used in conjunction with different filters to detect battery defects. Overall the processing steps are described in an easy to follow manner.
However, the details in experiment section to demonstrate the performance of the proposed approach shall be improved. The label 1-5 in table 1 and Methods label in table 2 are not clearly defined. The traditional method used as benchmark is also not discussed. Therefore it is hard to justify the benefit of the proposed approach as compared to traditional methods as claimed in the paper. The authors need to make this point stronger.
Also, as the author mentioned the used of the proposed method in actual production line, it would be good reference to elaborate limitation details of software to be used in real-world, e.g., the minimum speed requirement, the acceptable accuracy.
In introduction section, it is not clear that the work mentioned are used to detect what types of defects and whether there is any work attempting to detect battery defects as proposed in this work.
Round 2
Reviewer 1 Report
The authors have addressed my comments correctly.I have suggested to make the database available for the public and the author declared that "will submit my experiment data later" - I hope that the camera ready version of the paper will include the appropriate reference for the database. I will stress again that data sharing is very important in the scientific community.
Reviewer 2 Report
Thanks for addressing my comments.